# Real-Time Temperature Prediction of Power Devices Using an Improved Thermal Equivalent Circuit Model and Application in Power Electronics

**Zhen Hu** [1,*] , **Man Cui** [2,*] **and Xiaohua Wu** [3]

1   College of Automation, Nanjing University of Posts and Telecommunications, Nanjing 210023, China
2   School of Information and Electronics, Beijing Institute of Technology, Beijing 100081, China
3   School of Computer Science, Nanjing University of Posts and Telecommunications, Nanjing 210023, China; wuxiaohua@njupt.edu.cn
*   Correspondence: huzhen0111@njupt.edu.cn (Z.H.); 7520210140@bit.edu.cn (M.C.)

**Abstract:** As a core component of photovoltaic power generation systems, insulated gate bipolar transistor (IGBT) modules continually suffer from severe temperature swings due to complex operation conditions and various environmental conditions, resulting in fatigue failure. The junction temperature prediction guarantees that the IGBT module operates within the safety threshold. The thermal equivalent circuit model is a common approach to predicting junction temperature. However, the model parameters are easily affected by the solder aging. An accurate temperature prediction by the model is impossible during service. This paper proposes an improved thermal equivalent circuit model that can remove the effect of solder aging. Firstly, the solder aging process is monitored in real-time based on the case temperatures. Secondly, the model parameters are corrected by the thermal impedance from chip to baseplate based on the linear thermal characteristic. The simulation and experimental results show that the proposed model can reduce the temperature prediction error by more than 90% under the same aging condition. The proposed method only depends on the case temperatures to correct the model parameters, which is more economical. In addition, the experimental and simulation analysis in this work can help students of power electronics courses have an in-depth knowledge of power devices' mechanical structure, heat dissipation principles, temperature distribution, junction temperature monitoring, and so on.

**Keywords:** temperature prediction; power device; thermal model; reliability

## 1. Introduction

Photovoltaic power generation technology has taken a significant leap in the past several decades, due to advances in materials, power converters, and energy storage. Photovoltaic power generation systems usually exist in arid natural environments such as deserts. Various components easily break due to the complex operation environments. The photovoltaic power generation industry is cost-sensitive to the market, and frequent failures may make photovoltaic power generation lose competitiveness [1]. Therefore, new technologies are necessary to improve the performance and reliability of the system during operation. Power converters consist of IGBT modules, which are one of the most vulnerable components in the system. Due to harsh environments and unpredictable mission profiles, IGBT modules are often subjected to enormous and uncertain temperature fluctuations, leading to degradation of electrical performance and even device damage. Studies show that temperature caused more than 60% of device failures [2]. In addition, the probability of device failure doubles for every 10 °C increase in temperature [3]. Therefore, accurately predicting the temperature of the IGBT modules and making the modules run below the safety threshold is a vital means to improve the reliable performance of the device, and it is also a guarantee for the development of the photovoltaic power generation industry.

Real-time junction temperature prediction is the key to extending the reliability of the IGBT modules. The thermal equivalent circuit model is a frequently used temperature prediction approach for long-term load scenarios because of its simplicity and high efficiency. In the past few years, many researchers have improved the performance of thermal equivalent circuit models in many aspects, such as thermal coupling between different layers or adjacent chips, thermal boundary states, temperature-dependent materials, and computational efficiency [4–9]. For instance, a lot of research in the literature uses the finite-element analysis method for detailed three-dimensional temperature information to solve the issue of thermal coupling and thermal boundary conditions. In addition, much work has been carried out on computationally efficient thermal behavior modeling for power semiconductor devices when the devices are healthy [10–13]. A temperature characteristics-based dynamic model is able to remove the effect of temperature on material properties [14–17]. Scognamillo et al. proposed an innovative technique that allows the experimental extraction of the junction-to-ambient thermal impedance ($Z_{TH}$) of power devices operating in their application environment [18]. The above-proposed methods improve the performance of thermal equivalent circuit models in different applications.

Unfortunately, the aforementioned means ignore the effect of solder aging on the model parameters. The solder aging changes the thermal path inside the IGBT modules, resulting in a mismatch between the model parameters and the device's mechanical structure [19–21]. The junction temperature prediction based on the model may be much lower than the real chip temperature, resulting in an optimistic evaluation of the module's operation conditions. Finally, the aging process of the IGBT module is accelerated. Based on the above analysis, a thermal equivalent circuit model that has the ability to remove the effect of solder aging is more necessary. In [22,23], model parameters were revised by the variations of thermal impedance due to solder aging. However, these methods still have some limitations: (a) Only the thermal resistance is corrected, and the heat capacity is neglected, due to the lack of research on the influence mechanism of solder aging on model parameters, resulting in the prediction accuracy of junction temperature cannot be completely restored. (b) The calculation of thermal impedance depends on the measurement of chip junction temperature. However, the temperature-sensitive electrical parameters (TSEP) including collector–emitter on-state voltage $V_{ce,on}$ and gate-source voltage $V_{gs}$ are susceptible to the bond wires and gate oxide degradation [24–26]. These limitations make it impossible to apply the model to high-precision situations. Therefore, there are still some challenges in correcting the model parameters completely and estimating the thermal impedance economically.

Motivated by the above analysis, this paper proposes an improved thermal equivalent circuit model that can remove the influence of solder aging on temperature prediction. This work includes two aspects: (a) the solder aging process is real-time monitored based on case temperatures and the thermal impedance variations are estimated according to the case temperatures; (b) model parameters consisting of thermal resistance and thermal capacitance are corrected based on linear thermal characteristics. The model parameters are corrected timely to obtain the device's accurate junction temperature through the above two stages. Finally, the reliability of power converters under various operational states is improved.

Power electronics is a theoretical course that introduces the basic principles, analytical methods, and typical applications of power electronics technology, which is the basis of professional knowledge in electrical disciplines. Power electronics contains the characteristics and use methods of semiconductor power devices, and the working principles and analysis methods of three-phase rectifier, chopper, inverter, and frequency conversion circuits. Structural characteristics and thermal reliability monitoring of semiconductor power devices are hardly involved. As the core component of a photovoltaic power generation system, the thermal reliability of the semiconductor power device affects the reliability of the whole power generation system. The thermal failure principle of semiconductor power devices and how to carry out thermal management are significant methods to improve stu-

dents' understanding of power devices. This paper introduces the temperature monitoring method and the thermal failure principle of the power device. At the same time, the finite element analysis method and accelerated aging test method are introduced in the simulation and experiment section. Students can have a more comprehensive understanding of the mechanical structure, failure principle, heat dissipation mechanism, temperature monitoring, and so on.

The remainder of this article is as follows. In Section 2, the monitoring algorithm of solder aging in real-time and the correction theory of model parameters are introduced. In Sections 3 and 4, the finite-element analysis and experimental analysis are employed to validate the effectiveness of the proposed method, respectively.

## 2. Method

### 2.1. Monitoring of Solder Aging in Real-Time

The chip is considered the heat source of the IGBT module. The heat is generally generated on the upper surface of the chip and transferred from the chip to the baseplate through various layers with different materials. A great amount of heat flow spreads down with an angle of $45°$, resulting in the baseplate containing a higher temperature distribution than the chip, as shown in Figure 1.

The thermal impedance from chip to baseplate $Z_{JC}$ is a commonly used parameter to characterize the property of the thermal path. The formula of $Z_{JC}$ is as follows.

$$Z_{JC} = \frac{T_J - T_{C-chip}}{P} \tag{1}$$

where $P$ denotes the device's power loss, $T_J$ is the junction temperature of the chip, and $T_{C-chip}$ is the case temperature at the center of the baseplate.

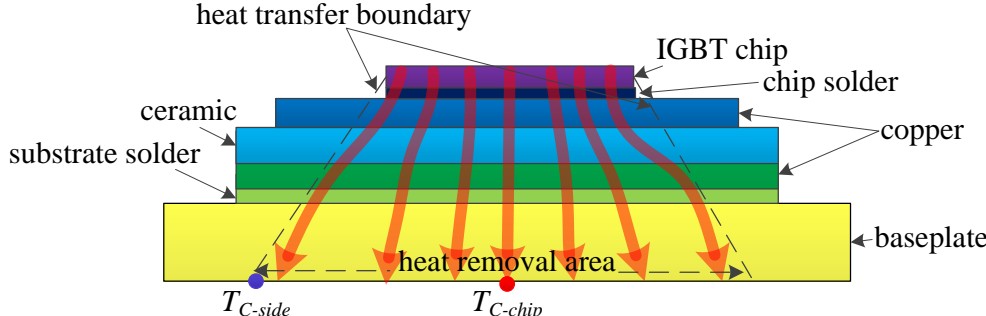

**Figure 1.** The thermal path of a healthy IGBT module (the arrows represent the heat flow).

A smaller value of $Z_{JC}$ results in faster heat diffusion. Solder aging originates from the layer's side region and extends to the central region, as shown in Figure 2. Solder aging indicates that cracks or voids are generated in the solder layer. The heat transfer channel becomes narrower due to the cracks and voids. Most heat can only be transferred to the baseplate by the central non-crack area, resulting in heat accumulating inside the module. The final expression is the increase of $Z_{JC}$ value. Because of this, $Z_{JC}$ is able to monitor solder aging timely. However, the $Z_{JC}$ estimation depends on the junction temperature $T_J$. The thermal equivalent circuit model cannot acquire the accurate junction temperature as the solder aging exists. As a result, it cannot accurately calculate $Z_{JC}$ by the model. The temperature-sensitive electrical parameters (TSEPs) method is another way to measure the junction temperature. However, the TSEPs method requires a high-precision circuit that is expensive. In addition, the TSEPs are susceptible to bond wires and gate oxide degradation. Therefore, a cost-effective and reliable monitoring method for solder aging is urgently needed.

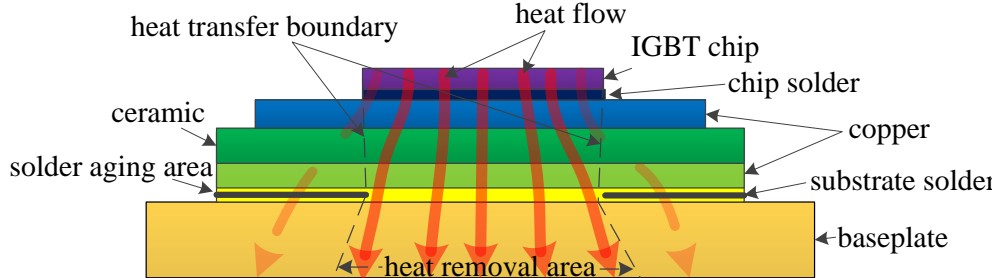

**Figure 2.** The thermal path of an aged IGBT module (the arrows represent the heat flow).

The baseplate's thermal diffusion property can be characterized by $Z_{CA}$ which is the thermal impedance from the baseplate to the ambient. The formula of $Z_{CA}$ is as follows.

$$Z_{CA} = \frac{T_C - T_A}{P} \tag{2}$$

where $P$ is the device's power loss, $T_C$ is the case temperature of the baseplate, and $T_A$ is the ambient temperature.

Since the temperature distribution of the baseplate is nonuniform, the thermal diffusion properties of different regions are distinct. That is to say, the value of $R_{CA}$ is various. Most heat generated in the chip transfers to the baseplate along the optimal path, i.e., the vertical direction. Therefore, the central temperature of the baseplate is higher than the rest. Solder aging generally originates from the side area and extends to the central area. As a result, the heat accumulates in the central area of the baseplate when the solder aging occurs. The temperature non-uniformity of the baseplate is intensified. Finally, $R_{CA}$ in the central area increases, and $R_{CA}$ in the rest decreases.

There are two case temperatures that are susceptible to the solder fatigue. One is $T_{C-chip}$, i.e., the central point of the baseplate. The other is $T_{C-side}$, i.e., the origin point of solder aging. $Z_{CA}$ of the above two points is estimated by (2), respectively.

$$Z_{CA-chip} = \frac{T_{C-chip} - T_A}{P}$$
$$Z_{CA-side} = \frac{T_{C-side} - T_A}{P} \tag{3}$$

where $Z_{CA-chip}$ denotes the thermal impedance of the central point in the baseplate, and $Z_{CA-side}$ is the thermal impedance of the side point in the baseplate. With the solder aging process, $T_{C-chip}$ continues to increase, while $T_{C-side}$ decreases. As a result, $Z_{CA-chip}$ grows, and $Z_{CA-side}$ becomes smaller. Therefore, $Z_{CA-chip}$ or $Z_{CA-side}$ is also able to monitor solder aging process timely. Compared with $Z_{JC}$, $Z_{CA-chip}$ and $Z_{CA-side}$ independent of junction temperature are acquired at a lower cost. Considering that $Z_{CA-chip}$ and $Z_{CA-side}$ also depend on the module's power loss, this paper proposes a new parameter $k_p$ to monitor the solder aging. The formula of $k_p$ is as follows.

$$k_p = \frac{Z_{CA-chip}}{Z_{CA-side}} = \frac{T_{C-chip} - T_A}{T_{C-side} - T_A} \tag{4}$$

From (4), the $k_p$ value continues to increase with the degradation of solder layer. The value of $k_p$ only depends on the case temperatures, and is independent of the device's power loss and the junction temperature. Therefore, it has a better economy and stability with the $k_p$ parameter to monitor the solder aging. An offline accelerated aging test of power devices can set up a database consisting of $k_p$ and $Z_{JC}$. In practice, $Z_{JC}$ is acquired from the $k_p$ value for quantitative evaluation of solder aging. In addition, the change in $Z_{JC}$ is used to correct the parameters of the thermal equivalent circuit model.

### 2.2. Online Correction of the Model Parameters

The thermal equivalent circuit model parameters contain thermal resistance $R_i$ and thermal capacitance $C_i$. $R_i$ parameter relates to the model's steady-state characteristics, and $C_i$ parameter relates to the model's dynamic characteristics. The definitions of $R_i$ and $C_i$ are as follows.

$$R_i = \frac{d_i}{\lambda_i \cdot A_i} \tag{5}$$

and

$$C_i = c_i \cdot \rho_i \cdot d_i \cdot A_i \tag{6}$$

where $d_i$ and $A_i$ are the thickness and heat surface area of the $i$th layer, respectively; $\lambda_i$, $c_i$ and $\rho_i$ are the thermal conductivity, specific heat capacity, and density of the material for the $i$th layer, respectively.

From (5) and (6), both $R_i$ and $C_i$ are related to the heat surface area of each layer. Solder aging reduces the heat surface area of the solder layer. Therefore, $R_i$ and $C_i$ parameter based on the initial heat surface area are no longer suitable for junction temperature prediction. $Z_{JC}$ is a significant parameter to characterize the module's thermal path. Solder aging changes the thermal path of the module, as shown in Figure 2. The final result of the thermal path change is a variation in $Z_{JC}$. In other words, the change in $Z_{JC}$ reflects the change in the heat surface area of the solder layer, i.e., the thermal path. The change in heat surface area cannot be measured online due to the enclosed package. Considering the module's linear thermal characteristics, the proportion of change in heat surface area is consistent with the proportion of change in $Z_{JC}$.

According to the module's linear thermal behavior, the changes in heat surface area can be mapped proportionally to the model parameters. $R_i$ can be corrected based on the following formula.

$$R_{i(aged)} = R_i\left(1 + \frac{\triangle Z_{JC}}{\sum_{i=1}^{n} R_i}\right) \tag{7}$$

where $R_{i(aged)}$ denotes the aged thermal resistance, $\triangle Z_{JC}$ is the change of $Z_{JC}$, representing the change in heat surface area. The change in heat surface area due to solder aging is proportionally assigned to each thermal resistance parameter through (7). As a result, the influence of solder fatigue on thermal resistance parameters is removed. Besides, there is a time-constant parameter $\tau_i$ to characterize the model's thermal behavior. $\tau_i$ is composed of thermal resistance $R_i$ and thermal capacitance $C_i$. The formula of $\tau_i$ is as follows.

$$\tau_i = R_i \cdot C_i = \frac{c_i \cdot \rho_i \cdot d_i^2}{\lambda_i} \tag{8}$$

From (8), the $\tau_i$ value is independent of the heat surface area. That is, $\tau_i$ remains constant when the solder aging occurs. Therefore, $C_i$ is corrected by substituting $R_{i(aged)}$ into (8).

$$C_{i(aged)} = \frac{\tau_i}{R_{i(aged)}} \tag{9}$$

where $C_{i(aged)}$ is the aged thermal capacitance. From (7) and (9), the key to correct the model parameters is the change of $Z_{JC}$ due to the solder aging. In Section 2.1, we proposed a $k_p$ parameter to monitor the solder fatigue in real-time and to obtain the $Z_{JC}$ value. Therefore, compared with other methods, the proposed method in this paper only depends on the case temperatures to complete the correction of the model parameters.

### 2.3. Method Implementation

The implementation of the method includes two stages. One is the real-time monitoring of solder aging, and the other is the online correction of model parameters. In the first

stage, the case temperatures are measured in real time by thermal sensors placed between the baseplate and the heat sink. The layout of the sensors is in Figure 3.

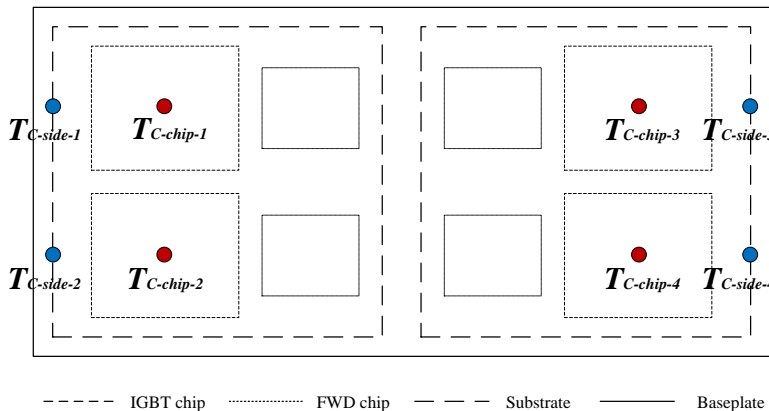

**Figure 3.** Placement of thermal sensors for measuring case temperatures.

The measured case temperatures are adopted to estimate the $k_p$ value through (4). The solder aging is decided by observing whether the $k_p$ value changes. When the solder aging occurs, the $Z_{JC}$ value is acquired through the updated $k_p$ value. In the second stage, substituting the change of $Z_{JC}$ into (7) and (9) to correct the model parameters. Then the junction temperature is predicted by the updated thermal equivalent circuit model. With the above two stages, the thermal equivalent circuit model is corrected timely during the solder aging. The flowchart of the method implementation is in Figure 4.

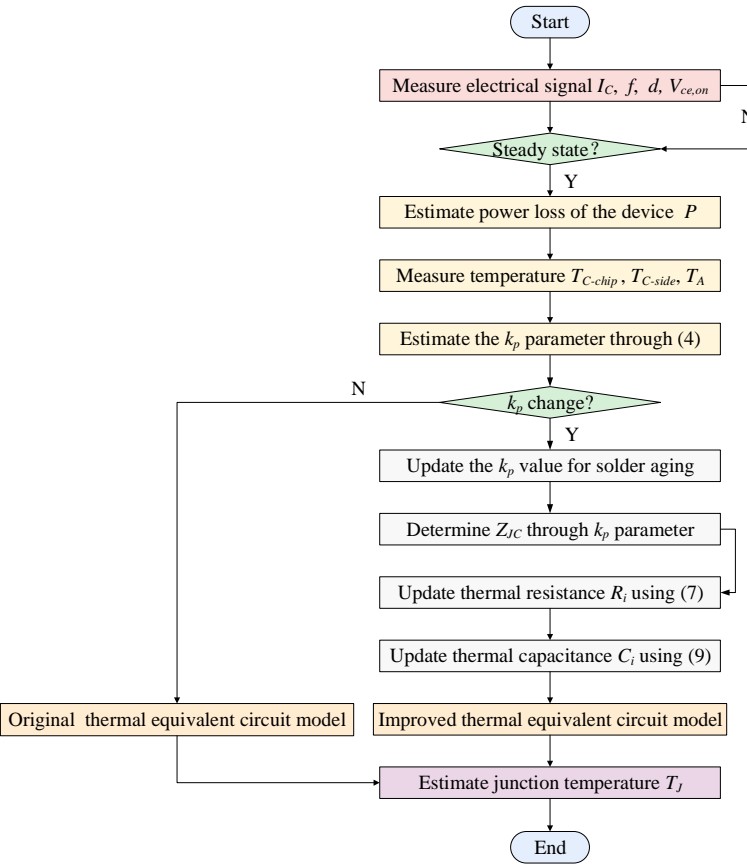

**Figure 4.** The flowchart of the method.

### 3. Simulation Validation

In this section, a finite element analysis (FEA) example is demonstrated to validate the effectiveness of the proposed method. A commercial IGBT module produced by SEMIKRON (Shanghai, China) is modeled through Pro/Engineer software (Version 5.1), as shown in Figure 5. The IGBT model shown in Figure 5 is introduced in ANSYS, a commercial FEA software platform (Version 17.2). Transient thermal analysis of the IGBT model is carried out in ANSYS software (Version 17.2). The operation conditions of the IGBT module are as follows: the DC-link voltage is 300 V, the collector current is 60 A, the switching frequency is 10 kHz, the modulation index is 1, and the line frequency is 50 Hz.

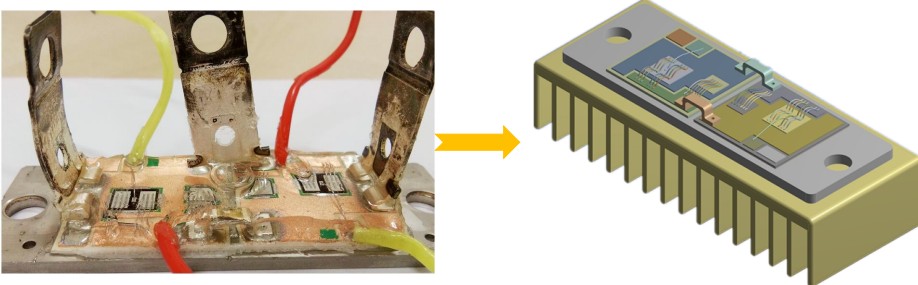

**Figure 5.** The three-dimensional IGBT model.

The solder fatigue is simulated by changing the thermal conductivity of the partial region in the solder layer. We set up the following aging scenarios to explore the effect of solder fatigue on the module's thermal behavior. (1) healthy condition without solder aging; (2) slightly aging, i.e., 10% aging region in the solder layer; (3) minor aging, i.e., 20% aging region in the solder layer; (4) intermediate aging, i.e., 30% aging region in the solder layer; (5) extensive aging, i.e., 40% aging region in the solder layer; (6) dangerous aging, i.e., 50% aging region in the solder layer. In addition, it should be noted that the solder aging gradually expanded from the side to the center.

The power loss of the chip is estimated according to the operation conditions of the IGBT module. The transient thermal analysis is processed by applying the power loss on the IGBT models with various aging conditions. The heat flow results under different aging scenarios are in Figure 6. From Figure 6, the thermal channel of the heat flow spreading down to the baseplate gradually narrowed from scenarios 1 to 6. The heat flow accumulates in the central area of the solder layer, resulting in a continuous increase in heat flux density in the central area with the degradation of the solder. In scenario 6, the heat flow can only be transferred down to the baseplate through the non-cracked area in the solder layer. Therefore, the heat flow is concentrated in the baseplate's central area, while reduced in the remaining area. Finally, the case temperatures in the central region of the baseplate increase, while the case temperatures in other regions continue to decrease.

The evolution of $T_{C-chip}$ and $T_{C-side}$ during the degradation of solder layer is in Figure 7. $T_{C-chip}$ gradually increases about 5 °C from scenario 1 to 6. Meanwhile, $T_{C-side}$ continues to decrease by about 2.5 °C. $T_{C-chip}$ and $T_{C-side}$ are substituted into (4) to estimate the $k_p$ values under different solder aging conditions. In addition, $Z_{JC}$ values are estimated by substituting $T_J$ and $T_{C-chip}$ into (1). The results of $k_p$ and $Z_{JC}$ are in Figure 8.

With the deterioration of solder aging, the values of $k_p$ and $Z_{JC}$ increase monotonically, indicating that $k_p$ and $Z_{JC}$ are only related to the solder aging. Therefore, a database including $k_p$ and $Z_{JC}$ is built based on an offline accelerated aging test. In practice, the solder fatigue is monitored in real time by the $k_p$ parameter. The $Z_{JC}$ value is acquired through $k_p$ in the database. Compared with other methods, the $k_p$ parameter only relies on the case temperatures, with an advantage in economy and stability.

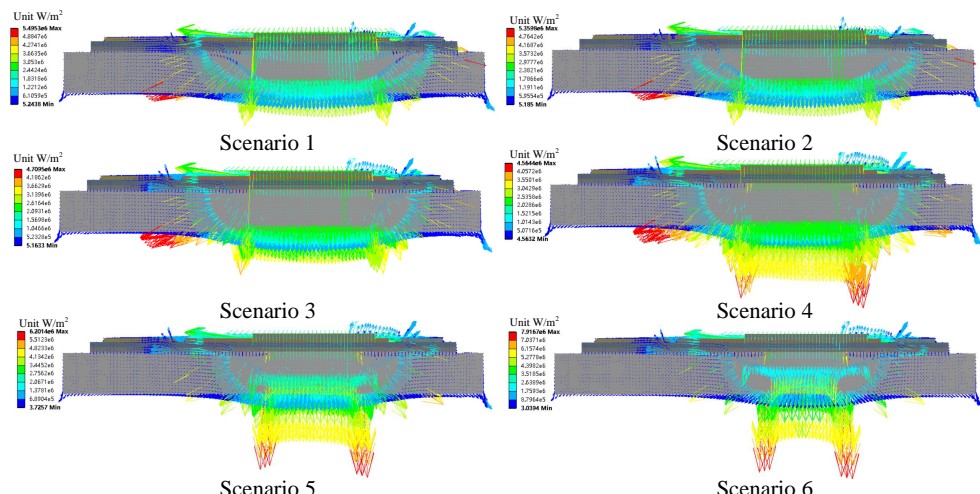

**Figure 6.** The heat flow under various solder fatigue states (the arrows in different colors represent various heat fluxes).

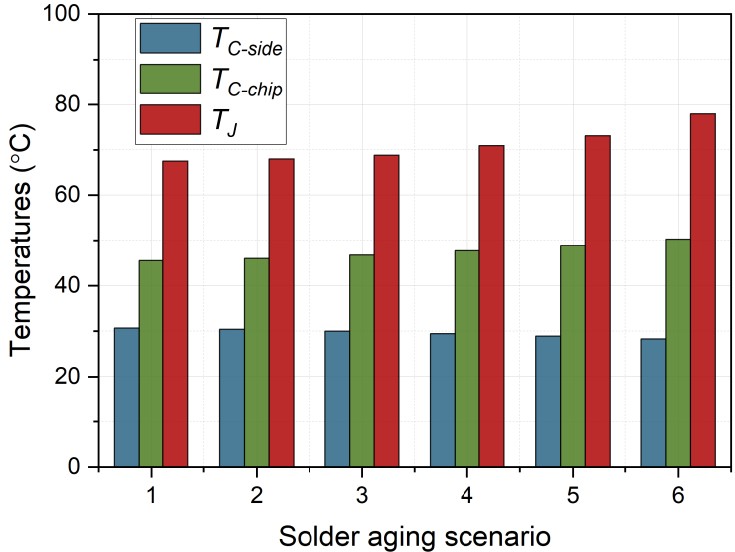

**Figure 7.** Evolution of temperatures during solder fatigue.

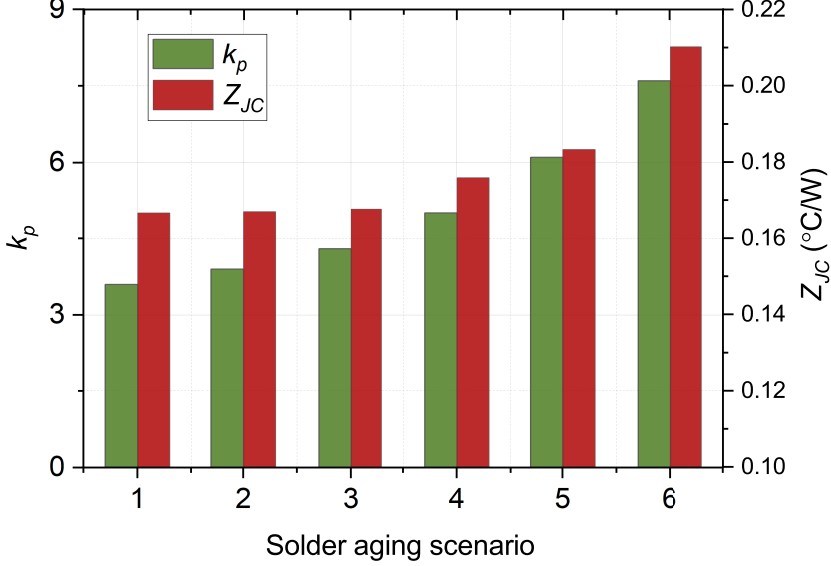

**Figure 8.** Evolution of $k_p$ and $Z_{JC}$ during solder fatigue.

Then, the effectiveness of the correction algorithm for the model parameters is verified. The parameters of the thermal equivalent circuit model for a healthy IGBT module are extracted based on the work in [27,28]. The extracted model parameters are in Table 1. Firstly, the performance of the original thermal equivalent circuit model during solder aging is tested. The operation condition and case temperature in scenario 5 are given to the original thermal equivalent circuit model to predict junction temperature. The junction temperature results from the original model and FEA are compared in Figure 9.

**Table 1.** Parameters of the original thermal equivalent circuit model.

| $i$ | 1 | 2 | 3 | 4 |
|---|---|---|---|---|
| thermal resistance $R_i$ | 0.014 | 0.0435 | 0.0732 | 0.0358 |
| thermal capacitance $C_i$ | 16.55 | 0.2175 | 0.487 | 0.032 |

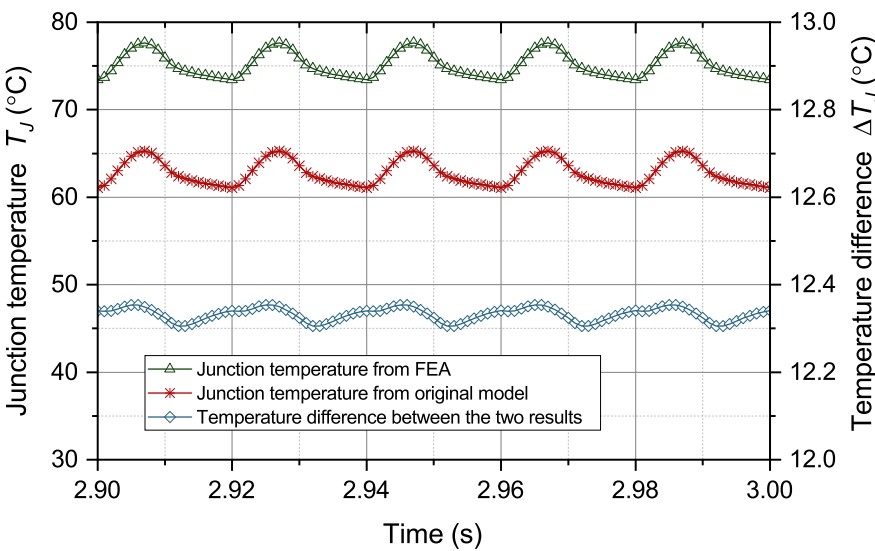

**Figure 9.** Temperatures from FEA and the original model.

The difference between the two temperature results is greater than 12 °C most of the time, which greatly exceeds the tolerable error of the temperature prediction. Continued use of this model may cause the module to operate outside the safety threshold. Therefore, it is necessary to correct the model parameters to remove the influence of solder aging. The model parameters are corrected through (7) and (9) based on the information of $Z_{JC}$. The modified model parameters are in Table 2.

The results in Table 2 show that the value of $R_i$ increases gradually with the deterioration of solder aging, while the value of $C_i$ continues to decrease. Different material properties of the crack and the solder caused this phenomenon. Compared with solder, the thermal conductivity of the crack is lower, and the specific heat capacity is greater. A low thermal conductivity increases $R_i$, and a high specific heat capacity decreases $C_i$.

**Table 2.** Modified parameters of the improved model for the six simulated aging scenarios.

| Parameters | $R_1$ | $R_2$ | $R_3$ | $R_4$ | $C_1$ | $C_2$ | $C_3$ | $C_4$ |
|---|---|---|---|---|---|---|---|---|
| scenario 1 | 0.014 | 0.0435 | 0.0732 | 0.0358 | 16.55 | 0.2175 | 0.487 | 0.032 |
| scenario 2 | 0.01403 | 0.0436 | 0.07335 | 0.03587 | 16.517 | 0.217 | 0.486 | 0.0319 |
| scenario 3 | 0.01409 | 0.0438 | 0.0736 | 0.036 | 16.45 | 0.216 | 0.484 | 0.0318 |
| scenario 4 | 0.0148 | 0.046 | 0.077 | 0.0378 | 15.66 | 0.206 | 0.46 | 0.030 |
| scenario 5 | 0.0154 | 0.0479 | 0.08 | 0.039 | 15.03 | 0.198 | 0.44 | 0.029 |
| scenario 6 | 0.0177 | 0.055 | 0.092 | 0.045 | 13.1 | 0.172 | 0.386 | 0.025 |

The same operation conditions are applied to the improved thermal equivalent circuit model to predict the junction temperature. The comparison of the junction temperature results from the model and FEA is in Figure 10.

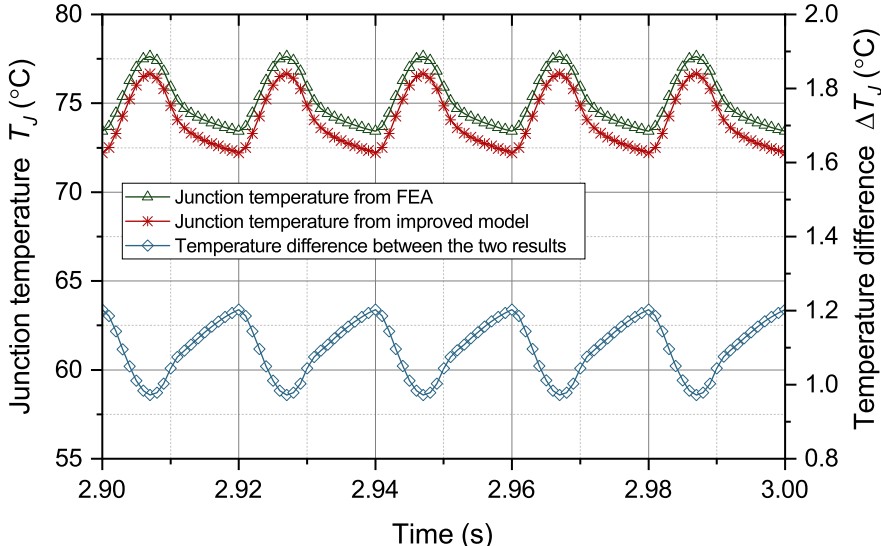

**Figure 10.** Temperatures from FEA and the improved model.

From Figure 10, the temperature results from the improved thermal equivalent circuit model can accurately track the temperature results from FEA. The difference between the two temperature results was generally less than 1.2 °C. The correlation between the two signals is more than 0.96. Under the six solder aging scenarios, the mean absolute error (MAE) of junction temperature from the original and improved thermal equivalent circuit models is in Table 3. From Table 3, the temperature prediction error from the original model continues to increase with the deterioration of the solder. Under scenario 6, the prediction error of the original model reaches 32 °C, which is far beyond the tolerance of temperature prediction. Compared with the original model, the improved model can significantly reduce the prediction error by about 90%. As a result, the performance of temperature prediction greatly improved.

**Table 3.** Error Statistics of $T_J$.

| Model Type | Original Model | Improved Model |
|---|---|---|
| Error in scenario 1 (°C) | 0.016 | \ |
| Error in scenario 2 (°C) | 0.24 | 0.02 |
| Error in scenario 3 (°C) | 0.75 | 0.07 |
| Error in scenario 4 (°C) | 6.9 | 0.6 |
| Error in scenario 5 (°C) | 12.33 | 1.1 |
| Error in scenario 6 (°C) | 32 | 2.9 |

## 4. Experimental Validation

In this section, the effectiveness of the improved thermal equivalent circuit model is validated by an experimental case. The test equipment is in Figure 11, including a commercial IGBT module produced by SEMIKRON (the upper package is removed), an IR camera to measure the chip junction temperature, a recorder for obtaining various electrical signals of the module, a signal generator for offering a driving signal, a DC power supply for the test current, air-cooled equipment to cool the module, and a National Instruments (NI) data acquisition instrument for measuring the case temperatures. Shallow grooves are carved into the upper surface of the heat sink for placing thermal sensors to measure the case temperatures.

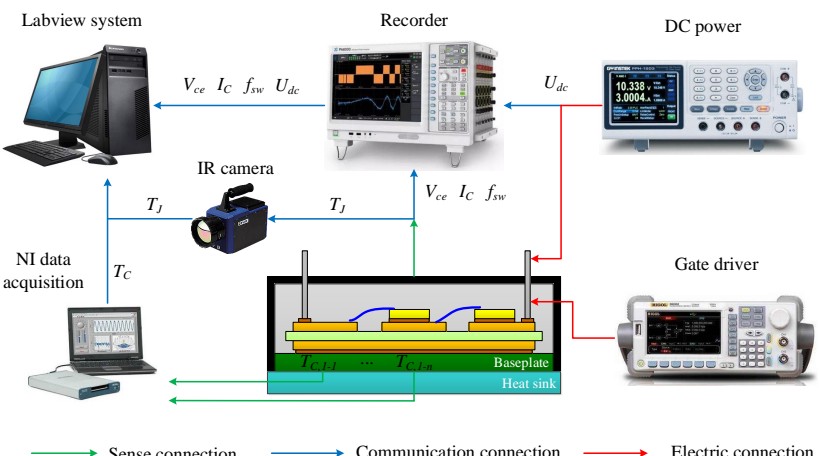

**Figure 11.** Experimental setup.

To obtain the evolution of various signals that characterize the IGBT module's aging conditions, we built an accelerated aging test platform for power devices based on the experimental equipment shown in Figure 11. Applying a 10% overload current to the IGBT module to heat the junction temperature to 180 °C quickly, and then bring down the junction temperature to 60 °C through the air-cooled equipment. In this way, the IGBT module is subject to a 120 °C temperature swing in one thermal cycle.

The starting point of the thermal cycle in Figure 12 is 0, meaning that the IGBT module is a healthy device without thermal damage. Thermal damage occurs to the solder and bond wires when the IGBT module goes through a thermal cycle. The IGBT fails when the thermal damage accumulates to the threshold value. The $k_p$ and $Z_{JC}$ are the indicators of solder aging. The values of $k_p$ and $Z_{JC}$ rely on temperature information such as $T_{C-chip}$, $T_{C-side}$ and $T_J$. The case temperatures containing $T_{C-chip}$ and $T_{C-side}$ are collected in real time by the NI data acquisition instrument during the test, and $T_J$ is measured with the IR camera. The collector–emitter on-voltage $V_{ce,on}$ indicates the degradation of bond wires, and $V_{ce,on}$ is collected in real time by the recorder. $T_{C-chip}$, $T_{C-side}$ and $T_J$ are substituted into (1) and (4) to estimate the values of $k_p$ and $Z_{JC}$, respectively. The results of $k_p$, $Z_{JC}$ and $V_{ce,on}$ are in Figure 12.

$V_{ce,on}$ is a core parameter to describe the bond wires' degradation. The thermal damage to the module containing solder and bond wires has a very slow growth before 3000 thermal cycles. Therefore, the values of $k_p$, $Z_{JC}$ and $V_{ce,on}$ have a little change. According to the theory of cumulative damage in fatigue, the module's thermal damage reaches the threshold after 3000 cycles. Accordingly, $k_p$, $Z_{JC}$ and $V_{ce,on}$ have a rapid growth after 3000 cycles. When one bond wire falls off, $V_{ce,on}$ increases exponentially, resulting in a big rise in the module's power loss. As a result, the thermal pressure of the solder layer increases, leading to accelerated solder aging. The growth trend of $k_p$ and $Z_{JC}$ is consistent, indicating that $k_p$ and $Z_{JC}$ are only related to solder aging. Therefore, the $k_p$ parameter is suited to monitor solder fatigue timely. According to the information of $k_p$, $Z_{JC}$ is acquired during the normal operation of power converter.

A DC pulse current is applied to the IGBT module to examine the performance of the original thermal equivalent circuit model during solder fatigue. The module's power loss is estimated according to the electrical signal collected by the recorder. The power loss is given to the original thermal equivalent model to predict the junction temperature. The temperature results from the model were compared with that measured by the IR camera, as shown in Figure 13. The IR camera used in this paper is the FOTRIC series, which can continuously collect temperature at a point and obtain a temperature map. Figure 14 shows the temperature distribution of the module's upper surface. From Figure 14, the temperature at the chip is the highest. The farther away it is from the chip, the cooler it becomes. That is because the chip is the heat source of the IGBT module.

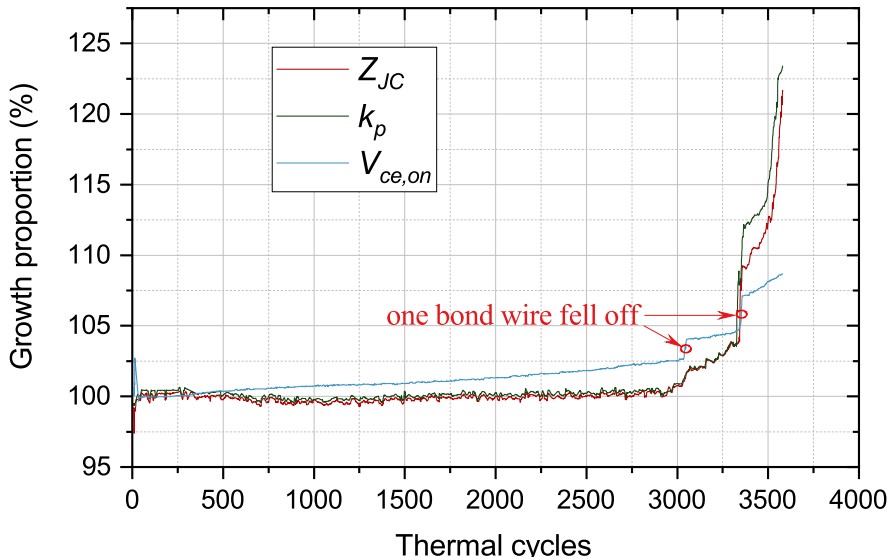

**Figure 12.** Evolution of $k_p$, $Z_{JC}$ and $V_{ce,on}$ during the solder fatigue.

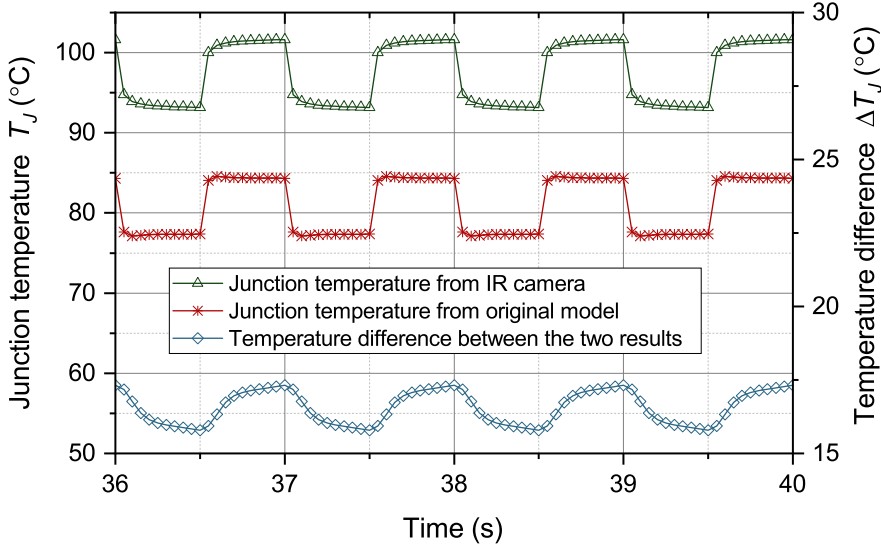

**Figure 13.** Temperatures from IR camera and the original model.

From Figure 13, the difference between the two temperature results is greater than 15 °C most of the time, with a maximum difference of 17 °C. Continued use of this model may cause the module to operate outside the safety threshold. Therefore, it is necessary to correct the model parameters to remove the effect of solder fatigue. The model parameters are corrected through (7) and (9) based on the information of $Z_{JC}$. The modified model parameters are in Table 4. From Table 4, it can be seen that the experimental results are consistent with the simulation results. The different material properties of the solder and crack lead to the variation in $R_i$ and $C_i$. The crack has lower thermal conductivity and higher specific heat capacity, resulting in an increase in $R_i$ and a decrease in $C_i$ with the deterioration of the solder.

The same operation conditions are applied to the improved thermal equivalent circuit model to predict the junction temperature. The comparison of the junction temperature results from the model and IR camera is in Figure 15. From Figure 15, the temperature results from the improved thermal equivalent circuit model can be seen to accurately track the temperature results from the IR camera. The difference between the two temperature results was generally less than 2 °C. The correlation between the two signals is more than 0.96. The mean absolute error (MAE) of junction temperature from the original and

improved thermal equivalent circuit models under the specified solder aging conditions is in Table 5. The simulation and experimental results are consistent. The junction temperature error of the original model increases with the solder aging. The junction temperature error of the original model reaches 16.2 °C when $Z_{JC}$ rises by 20%. It is no longer helpful for the thermal management of power devices. Compared with the original model, the junction temperature error from the improved thermal equivalent circuit model is reduced by about 90% under the specified aging conditions. As a result, the performance of temperature prediction greatly improved.

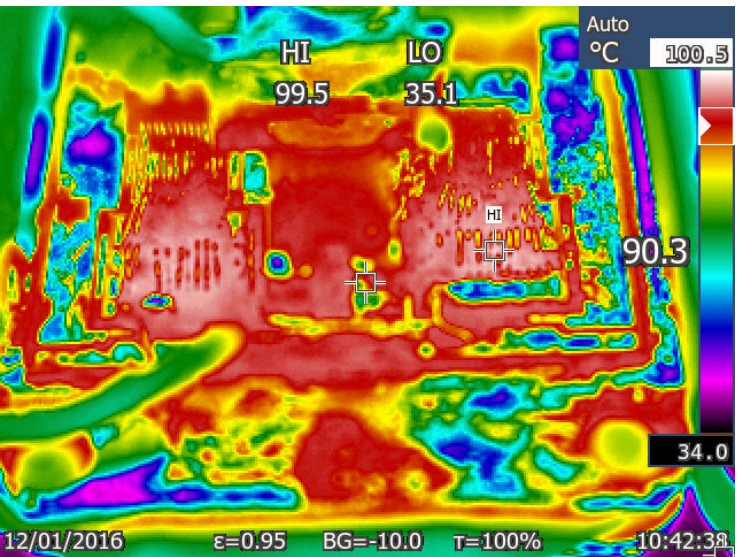

**Figure 14.** Temperature results from IR camera.

**Table 4.** Modified parameters of the improved model under the specific test aging conditions.

| Parameters | $R_1$ | $R_2$ | $R_3$ | $R_4$ | $C_1$ | $C_2$ | $C_3$ | $C_4$ |
|---|---|---|---|---|---|---|---|---|
| $Z_{JC}$ grows 0% | 0.014 | 0.0435 | 0.0732 | 0.0358 | 16.55 | 0.2175 | 0.487 | 0.032 |
| $Z_{JC}$ grows 5% | 0.0147 | 0.0457 | 0.0769 | 0.0376 | 15.76 | 0.207 | 0.464 | 0.030 |
| $Z_{JC}$ grows 10% | 0.0154 | 0.0479 | 0.08 | 0.0393 | 15.04 | 0.198 | 0.44 | 0.029 |
| $Z_{JC}$ grows 15% | 0.0161 | 0.05 | 0.084 | 0.041 | 14.39 | 0.189 | 0.423 | 0.0278 |
| $Z_{JC}$ grows 20% | 0.0168 | 0.052 | 0.088 | 0.043 | 13.79 | 0.181 | 0.406 | 0.0267 |

In addition, this work can help students of power electronics courses to understand the thermal failure mechanism of semiconductor power devices and how to improve the reliability of power device operation through temperature monitoring. The semiconductor power device is the core component of energy conversion. In power electronics courses, students only learn the principles of energy conversion through power devices. Most experimental courses are arranged around DC-AC conversion. Students cannot understand the internal structure of the power device, the heat generation mechanism, and the heat dissipation process of the power device. This experimental and simulation analysis can help students have an in-depth knowledge of power devices' mechanical structure, heat dissipation principles, temperature distribution, junction temperature monitoring, and so on. We suggest that this work can be included in the experimental courses of power electronics to give students a more comprehensive understanding of power devices.

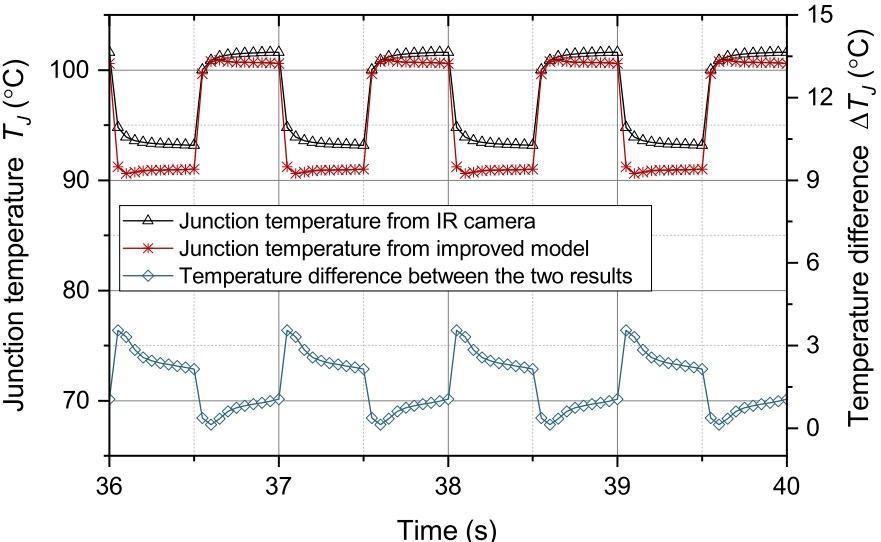

**Figure 15.** Temperatures from IR camera and the improved model.

**Table 5.** Error Statistics of $T_J$.

| Model Type | Original Model | Improved Model |
|---|---|---|
| Error in 0% growth of $Z_{JC}$ (°C) | 0.02 | \ |
| Error in 5% growth of $Z_{JC}$ (°C) | 4 | 0.42 |
| Error in 10% growth of $Z_{JC}$ (°C) | 8.2 | 0.85 |
| Error in 15% growth of $Z_{JC}$ (°C) | 12.3 | 1.3 |
| Error in 20% growth of $Z_{JC}$ (°C) | 16.2 | 1.6 |

## 5. Conclusions

This paper proposes an improved thermal equivalent model to predict the junction temperature of the IGBT module. The temperature accuracy from the model still satisfies the precision requirements when the solder aging occurs. Firstly, we propose a $k_p$ parameter to monitor solder aging in real-time. The $k_p$ parameter is only related to the case temperatures, and the $k_p$ parameter continues to increase with the solder aging. The database of $k_p$ and $Z_{JC}$ is established according to the accelerated aging test of power devices. During the normal operation of the IGBT module, the $Z_{JC}$ value is determined by the $k_p$ parameter. Secondly, we explore the mechanism of solder aging on the thermal equivalent circuit model parameters. An algorithm for modifying model parameters is proposed based on the module's linear thermal behavior. The changes in $Z_{JC}$ are mapped proportionally to the model parameters. Simulation and experimental results validate the effectiveness of the improved model. During the solder aging, the junction temperature prediction errors from the improved model are reduced by about 90% from the original model. The research of this paper plays a significant role in improving the reliability of photovoltaic power generation systems.

**Author Contributions:** Conceptualization, Z.H. and X.W.; methodology, Z.H.; software, Z.H.; validation, X.W.; formal analysis, M.C.; investigation, Z.H.; resources, Z.H.; data curation, M.C.; writing—original draft preparation, Z.H.; writing—review and editing, Z.H.; visualization, Z.H.; supervision, M.C.; project administration, Z.H.; funding acquisition, Z.H. All authors have read and agreed to the published version of the manuscript.

**Funding:** This work was funded by a grant from the National Natural Science Foundation (622030527) and Natural Science Research Project of higher education institutions in Jiangsu Province (22KJB470007).

**Data Availability Statement:** The data presented in this study are available on request from the corresponding author.

**Conflicts of Interest:** The authors declare no conflicts of interest.

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
