# Peer review of "Real-Time Temperature Prediction of Power Devices Using an Improved Thermal Equivalent Circuit Model and Application in Power Electronics"

_micromachines, doi:10.3390/mi15010063_

Round 1

Reviewer 1 Report

Comments and Suggestions for Authors

This manuscript describes the development of a thermal equivalent circuit model that can be implemented to reduce fatigue failure in photovoltaic power generation technology by improving real-time junction temperature prediction. Previous models ignored the contributions of the effects of solder aging, which leads to an overly optimistic evaluation of operation conditions and, therefore further accelerating the aging process. The manuscript reports on 1) monitoring the solder aging process in real-time based on case temperatures and 2) corrections to model parameters to more accurately obtain the junction's temperature. This is a very well-written manuscript that provides an excellent explanation of the concepts and the model to a non-expert. After publication, I would strongly consider using this article for instruction in a course I teach. I have only very minor suggestions for improvement:

1. Figures 7 and 8 have an x-axis that corresponds to discrete scenarios, but the data points are connected by lines indicating that there can be interpolation between the scenarios. Either the x-axis should be something measurable (such as area or path distance) on a continuous scale, or  a different type of plot should be used.

2. The text on the plot in Figure 12 should be corrected to "one bond wire fell off"

Comments on the Quality of English Language

The quality of the English language is quite high throughout.

Reviewer 2 Report

Comments and Suggestions for Authors

GENERAL COMMENT:

The Authors proposed a valuable idea to estimate the solder degradation in power modules with electronic devices (IGBTs). More specifically, they have shown how the solder degradation can be measured as a shrinking of the cone representing the heat flux from the source i.e., the die) to the cooling surface (i.e., the bottom of the baseplate). I appreciate the idea, but I believe it is not well explained; if the reader is within the research of thermal modeling, he/she may understand; whereas, for people which are not expert of these topics (but might make use of the technique), the manuscript is really hard to read. Therefore, I recommend a major revision to improve the paper quality. I prepared a long list of comments below to ease the Authors it the improvement of the paper quality. Once the Author fulfill the point below (please, carefully), I would propose the acceptance of the manuscript.

SPECIAL REMARKS: 

-----

INTRODUCTION SECTION and REFERENCES: The reference list is poor and not supporting the presented work. The authors should have done a more appropriate literature review. The introduction section MUST be enriched with references; in the following lines, I gave you some hints about it.

Rows 26-27: "Studies show that temperature caused more than 60% of device failures" A reference would be useful.

Rows 35-38: "In the past few years, many [...]  temperature-dependence materials, and computational intensity[4–7]". You never cited papers from L. Codecasa, who is a guru in the modeling of electronic devices through model-order reduction procedure. Some suggestions (as DOI): (i) 10.1109/TCPMT.2019.2931465, (ii) 10.3390/en14154683.

Rows 38-41: "For instance, lots of research [...] computational density". This statement was never supported by references, despite the literature being populated by many examples like this about power modules (e.g.,  10.1109/TCPMT.2020.3007146). 

Rows 42-44: "The above-proposed [...] different application". You never cited novel techniques which are supposed to evaluate the thermal impedance of any electronic device in any environment by means of electrical measurement in the frequency domain: 10.1109/TPEL.2022.3174617.

Rows 46-48: "The solder aging [...] mechanical structure". Please provide some references for this statement.

Rows 58-62: "However, the temperature-sensitive [...] thermal impedance economically". Please provide some references for this statement.

-----

CORE OF THE MANUSCRIPT: A list of comments follows.

Equations (1), (2) and so on: The literature respect the following standard. The R us used to denote the thermal resistant, that is, the steady state value of the thermal impedance, which in turn is defined with the Z. I would recommend using the Z everywhere you talk about thermal impedance. A reference for this assumption is here: 10.1109/NEMO.2018.8503451.

Equation (4): It looks like a good idea, but please do not use "k" for defining this quantity; k is the thermal conductivity, and this quantity is not a conductivity (it is a measure of the solder layer degradation).

Figure 6: Is there a reason why you showed just 4 out of 6 scenarios? In addition, is the scale of the heat flux intensity (i.e., color and size) the same in the pictures (I don't think, but I may be wrong). A reader would enjoy from a color bar showing the intensity of the heat.

Table 2: It is not clear the scenario in which those data are calculated. It should be interesting to see the effect on single Ris and Cis in the remaining 5 scenarios (the #1 is in Table 1 and is correctly used as a reference).

Table 3: also here, the statistic error was evaluated only for scenario #5. As you already had Ansys simulations and as the analytical correction trough k (which, I repeat, is an unfortunate nomenclature for people working in thermal analysis) is fast, why don't you add the error for ANY scenario? Is the error growing up with the degradation level? Also in scenario #6, is the error acceptable?

Figure 12: it is absolutely hard-to-read. It is not this simple to get information about the relation about the quantities. At least, the reader has to know what's the starting point of the quantity at Thermal Cycle #0 (as the power cycling starts).

Figure 14: it looks as it is set there and not commented. In the present status it does not contain information. A useful way to give information is to show at different time instants. In addition, you should support the reader to identify is a simpler way the power module.

Rows 273-275: "The temperature results from the model were compared with that measured by the IR camera, as shown in Figure 13." A possible solution for the practical application of the method is the use of thermocouples on the bottom of the module (between the baseplate and the heatsink). Did you try solutions like this in the experimental campaign? If yes, please provide details. If not, please explain why.

-----

CONCLUSIONS: It is too much focused on the "k parameter" and does not give a big picture of the manuscript. Please consider conclusions written as a summary of the work, which may be read without a full read of the core of the manuscript. In an extra-short way, conclusions should look like: we found a parameter giving us information about the solder degradation, we verified it works through numerical simulations, we proved that this technique is suitable for experiments.

-----

Comments on the Quality of English Language

The quality of the English language is not bad, but the way things are explained and presented must be improved (as I shown in my letter).

Round 2

Reviewer 2 Report

Comments and Suggestions for Authors

Dear Authors, I really appreciate your effort in improving the paper and accomplish my comments. Green light on my side. Wish you a good christmas